



# Demersal fishery Impacts on Sedimentary Organic Matter (DISOM): A global harmonized database of studies assessing the impacts of demersal fisheries on sediment biogeochemistry

Sarah Paradis[1], Justin Tiano[2], Emil De Borger[3,4], Antonio Pusceddu[5], Clare Bradshaw[6], Claudia Ennas[5], Claudia Morys[6], Marija Sciberras[7]

[1]Geological Institute, ETH Zürich, Zürich, Switzerland
[2]Wageningen Marine Research, Wageningen University and Research, PO box 68, 1970 AB, IJmuiden, the Netherlands.
[3]Royal Netherlands Institute for Sea Research (NIOZ), Department of Estuarine and Delta Systems, 5 PO box 140, 4400 AC, Yerseke, the Netherlands.
[4]Ghent University, Department of Biology, Marine Biology Research group, Krijgslaan 281/S8, 9000 Ghent, Belgium
[5]Department of Life and Environmental Sciences, University of Cagliari, Via T. Fiorelli, 1, 09126 Cagliari, Italy.
[6]Department of Environment, Ecology and Plant Sciences, Stockholm University, Stockholm, Sweden
[7]The Lyell Centre, Heriot-Watt University, Edinburgh, EH14 4AP, UK

*Correspondence to*: Sarah Paradis (sparadis@ethz.ch)

**Abstract.** Marine sediments are one of the largest carbon reservoirs on the planet and play a key role in the global cycling of organic matter. Bottom fisheries constitute the most widespread anthropogenic physical disturbance to seabed habitats, and this has prompted NGOs and governments to act on regulating mobile bottom contacting fishing gear. However, the scientific evidence of the effects of bottom trawling on sediment biogeochemistry are highly diverse and present contrasting results. Here we present a global harmonized dataset of 71 independent studies that assess the effects of demersal fisheries on sedimentological (i.e. grain size, porosity) and biogeochemical (i.e. organic carbon, phytopigments, nutrient fluxes) properties: Demersal fishery Impacts on Sedimentary Organic Matter (DISOM) database (Paradis, 2023; https://doi.org/10.3929/ethz-b-000634336). We establish protocols to report metadata that will allow a better comparison of the results in order to improve our understanding of the effects of bottom trawling on the seafloor on a global scale. With this harmonized database, we aim to allow researchers to explore the effects of demersal fisheries in variable environmental settings to deconvolve the effects of this disturbance and provide efficient management strategies.

## 1 Introduction

Demersal fishing is one of the most harmful anthropogenic activities in the ocean given its widespread occurrence, covering nearly all continental margins (Amoroso et al., 2018; Kroodsma et al., 2018), and increasing intensity over time (Watson and Tidd, 2018). Fishing with mobile bottom contacting fishing gear (e.g. trawls, dredges) alters the composition of benthic communities, with a general decrease in faunal biomass and biodiversity after disturbance, though effects vary in accordance to fishing intensity, type of fishing gear, sediment type, and biological traits of the resident community (e.g. mobility and





longevity) (Hiddink et al., 2017, 2019; Sciberras et al., 2018). In addition, the mixing and resuspension of sediment that results from dragging the gears across the seafloor has been shown to change the biogeochemical and physical characteristics of fishing grounds (Martín et al., 2014a; Oberle et al., 2018).

Results from trawling studies on sediment biogeochemistry are variable (Epstein et al., 2022). Certain studies observed that demersal fisheries decrease sedimentary organic matter (Hale et al., 2017; Mayer et al., 1991; Tiano et al., 2019), while others reported increases (Palanques et al., 2014; Pusceddu et al., 2005a; Sciberras et al., 2016), and some did not find statistically significant effects (Bhagirathan et al., 2010; Smith et al., 2000). Similarly, variable effects have been observed on the organic matter remineralization rates, with both increases (Paradis et al., 2019; Polymenakou et al., 2005) and decreases (Tiano et al.,

2019; Warnken et al., 2003) after fishing disturbance. The inconsistency in these results highlight that the effects of demersal fisheries on biogeochemical processes on the seafloor may be context- and site-specific, and caution must be taken when upscaling the effects to global demersal fishing grounds (Sala et al., 2021).

The wide array of studies assessing the effects of demersal fisheries have their own unique designs and sampling strategies, which could affect the observed outcomes if they aren't considered: manipulative experimental studies report the immediate

effect of sediment disturbance, whereas comparative studies tend to assess chronic effects of demersal fisheries. In addition, the environmental conditions of the study area such as sediment grain size influence the effect of sediment disturbance. Coarser grained sediments present in very hydrodynamic environments are generally characterized by low organic carbon contents, higher permeabilities and stronger hydrodynamic regimes that lead to faster remineralization rates when compared to finer grained muddy environments (Burdige, 2007; Huettel et al., 2014). In contrast, higher organic carbon contents are found in

muddy seafloors given their higher surface area (Keil et al., 1998; Mayer, 1994). In addition, fishing descriptors such as gear penetration depth, fishing intensity and frequency between disturbance events determine the effect of bottom fishing on benthic communities (Hiddink et al., 2017; Sciberras et al., 2018), as well as on sediment biogeochemistry (Depestele et al., 2019; Tiano et al., 2019, 2021). Similarly, different results may be obtained if the study is conducted in an area that has been chronically fished, due to the cumulative effects of sediment disturbance (De Borger et al., 2021).

In order to properly understand the effects of bottom trawling on sediment biogeochemistry in diverse sedimentary environments worldwide, a global harmonized dataset of the studies that investigate the effects of mobile bottom contacting fishing gear on sediment biogeochemistry is needed. The need for this global harmonized dataset is reflected by the growing attention of the impacts of demersal fisheries on sediment organic carbon stocks (Legge et al., 2020), and the proposal of variable management strategies (Black et al., 2022; Epstein and Roberts, 2022). This harmonized database will allow

researchers, managers, and policy makers to determine the biogeochemical effects of bottom trawling within a global context. Moreover, it will allow reproducible meta-analyses to be conducted from different perspectives, as well as train and validate models that upscale the biogeochemical impacts of bottom trawling on a global scale.

Here we present the first global dataset that compiles information of the sedimentological (i.e. grain size, porosity) and biogeochemical (i.e. organic carbon, phytopigments, nutrient fluxes) variables. Importantly, this database harmonizes metadata





of the different studies, given their inevitable influence in their findings. Finally, we also specify important metadata that should be reported in future studies to allow their inclusion in global meta-analyses.

## 2 Data collection

### 2.1 Literature review

A literature search was conducted in Web of Science and Scopus for publications that included any of the following keywords:
(trawling OR dredging), (sediment OR seafloor), (organic matter OR nutrient OR organic carbon). This search produced 195 studies that spanned from 1964 to 2022 (Table S1). A preliminary screening process was performed using the title and abstract to assess whether the articles studied the effects of sediment disturbance by demersal fishing gear on biogeochemical properties of the seafloor. This preliminary screening discarded 129 studies, either because they were not studying the impact of fishing (87), they were studying the impact of demersal fishing but not on the seafloor (16) or did not address its effects on sediment
biogeochemistry or sedimentology (12). Additional studies were discarded when they attributed demersal fishing as a possible explanation of their results but did not study its effects directly (12), or because the article could not be accessed (2) (Table S1). Only studies with empirical data obtained from sampling and consequent analyses of sediments exposed to demersal fisheries were included in the database. Hence, data obtained from modelling or manipulation experiments in the laboratory were excluded.

From the 66 publications that passed this preliminary screening, only the studies that complied with the following inclusion criteria were retained: i) studies sampled fished and control sites and/or before and after the fishing disturbance, or fishing grounds with different fishing intensities (e.g. gradient studies), ii) studies provided data of sediment geochemical variables (Table S2), iii) studies presented novel data. This last inclusion criterium was added because several studies belong to the same project (e.g. INTERPOL project (Polymenakou et al., 2005; Pusceddu et al., 2005a); RESPONSE project (Cartes et al., 2007;
De Juan and Cartes, 2011; Palanques et al., 2014)), and often present duplicate results that could introduce bias to statistical analyses. This second screening process led to 51 publications that were included in the database (Table 1).

If a publication presented data for multiple independent studies, it was separated into several independent studies. This was the case if a publication included different study designs (e.g., Brown et al., 2005), sampled in different geographical locations (e.g., Atkinson et al., 2011; McLaverty et al., 2020a; Muntadas et al., 2015; Rosli et al., 2016), sampled in different
environmental settings (e.g., Sciberras et al., 2017, 2016), or compared different fishing gears (e.g., Tiano et al., 2019). This led to 71 independent studies from 51 publications. Mean, measure of variability (i.e. standard deviation, standard error, confidence interval) and sample sizes of sedimentological (e.g., grain size, dry bulk density) and biogeochemical (e.g., organic carbon content, total nitrogen, oxygen consumption) variables were extracted from these 71 studies. When data were only presented as figures and not in tabular form, the authors were contacted to provide missing data, or the values were extracted
using the freely accessible WebPlotDigitizer software (Rohatgi, 2022).





**Table 1. References of publications included in DISOM. Additional information of the publications that were screened prior to their inclusion in DISOM is given in Table S1.**

| Reference | Reference (continued) | Reference (continued) |
|---|---|---|
| Tsikopoulou et al., 2022 | Hale et al., 2016 | Brown et al., 2005 |
| Tiano et al., 2022 | Sciberras et al., 2016 | Polymenakou et al., 2005 |
| Lamarque et al., 2021 | Muntadas et al., 2015 | Pusceddu et al., 2005a |
| Morys et al., 2021 | Costa and Netto, 2014 | Trimmer et al., 2005 |
| Paradis et al., 2021 | Dannheim et al., 2014 | Sheridan and Doerr, 2005 |
| Ferguson et al., 2020 | Goldberg et al., 2014 | Falcão et al., 2003 |
| McLaverty et al., 2020a | Meseck et al., 2014 | Stone et al., 2003 |
| McLaverty et al., 2020b | Martin et al., 2014b | Warnken et al., 2003 |
| Silveira et al., 2020 | Palanques et al., 2014 | Fiordelmondo et al., 2003 |
| Ramalho et al., 2020 | Pusceddu et al., 2014 | Sparks-McConkey and Watling, 2001 |
| Rajesh et al., 2019 | Sañé et al., 2013 | Watling et al., 2001 |
| Paradis et al., 2019 | Atkinson et al., 2011 | Dolmer et al., 2001 |
| Tiano et al., 2019 | Liu et al., 2011 | Smith et al., 2000 |
| Ramalho et al., 2018 | Bhagirathan et al., 2010 | Tuck et al., 1998 |
| Van de Velde et al., 2018 | Sánchez et al., 2009 | Brylinsky et al., 1994 |
| Sciberras et al., 2017 | Simboura et al., 2008 | Eleftheriou et al., 1992 |
| Rosli et al., 2016 | Cartes et al., 2007 | Mayer et al., 1991 |

## 2.2 Data harmonization

Given the diverse configurations of individual studies and their context-specific outcomes, it is imperative to harmonize them
to facilitate the comparability of findings. Harmonization was conducted for the study designs, environmental factors, and
fishing descriptors.

Studies were separated into three main study designs: experimental, comparative control-impact, and comparative gradient
studies. Experimental studies are those where areas were experimentally disturbed using a specific fishing gear. These either
consisted of collecting samples within and outside an experimentally fished area (control-impact study, CI), before and after
an area was experimentally fished (before-after study, BA), or a combination of both (before-after-control-impact study,
BACI). Comparative control-impact studies were those where an area known to be fished and at least one undisturbed control
site with similar environmental conditions were sampled. Finally, comparative gradient studies collected samples in several
sites exposed to different fishing intensity to assess the effect of fishing effort on sediment biogeochemistry.

Categories of environmental factors such as habitat type (sediment type), and seasonality were standardized across all studies.
Two different habitat types were defined based on the sedimentological properties of the sites: "muddy" if the percentage of
mud (sediment grain size < 63 µm) was higher than that of sand, or "sandy" if the percentage of sand was higher than mud. If
this information was not available, we extracted this classification based on the manuscript's description, or from associated



bibliography of the study area. Seasons varied throughout the different climatic zones encompassed in this database. While temperate zones are characterized by four seasons (winter, spring, summer, fall), tropical zones usually have two main
contrasting seasons (wet and dry). Although local variations in seasons may exist, the dataset was harmonized into these different categories: winter, spring, summer, and fall, in temperate regions and wet, dry seasons in tropical regions.

Regarding the metadata associated to fishing activity, we included several parameters of interest: fishing gear type, fishing effort, time since first and last disturbance, and if the site had been historically fished. Fishing gear type was classified into six main categories: otter trawls, beam trawls, dredges, electric pulse trawl, artisanal fishing gears, and undefined fishing gears
(Table 2). If a study did not specify what type of fishing technique was used, it was extracted from other studies or technical reports from the area, and if this information could not be obtained anywhere, it was assigned as "undefined". Although this classification may mask the nuances in the degree of impact each specific fishing gear could have, it groups together gears with similar properties.

**Table 2. Demersal fishing gear types included in DISOM.**

| Gear type | Characteristics |
| --- | --- |
| Otter trawl | Net is held open by two boards (otter trawls) at its extremes, which provide greater flexibility to adapt to rougher and steeper terrain.[1] |
| Beam trawl | Net is held open by a horizontal bar (beam) above the seafloor and collect demersal fauna that live on the seafloor.[1] |
| Dredge | Net is held open by a horizontal metal bar or blade that scrape or dig the seafloor depending on the targeted commercial species.[1] |
| Electric pulse trawl | Electrodes produce electric stimuli that drive flatfish out of the sediment, which are then captured by the dragged net behind.[2] |
| Artisanal | Combination of artisanal gears (passive and active) at night using light to promote the capture of demersal fisheries[3] |
| Undefined | Undefined fishing gear type, usually a combination of otter trawls, beam trawls, and dredges. |

[1]Martín et al., 2014a; [2]Tiano et al., 2019; [3]Silveira et al., 2020


Fishing effort was defined inconsistently among studies: some reported fishing effort as density of fishing vessels (Trimmer et al., 2005), others as number of hauls per surface area (Paradis et al., 2021), or presence of trawl tracks using side-scan sonars (Muntadas et al., 2015). Whenever possible, we harmonized fishing effort to fishing frequency ($y^{-1}$), as adopted in several meta-analyses of fishing disturbance (Hiddink et al., 2017; Sciberras et al., 2018). This unit is equivalent to swept area ratio
($km^2 \cdot km^{-2} \cdot yr^{-1}$) often employed in fishery studies and is representative of the number of disturbances that the site is experiencing. In the case of experimental studies, we harmonized the data to number of times the entire experimental area was fished prior to sampling, considering only the known fishing disturbance by the experimental set up and ignoring fishing effort by commercial demersal fisheries in the area, if applicable. In the case of comparative control-impact and gradient studies, ~75 % of the studies reported fishing intensity as fishing frequency, and in the remaining cases it was calculated multiplying
the number of hours fished by the speed and the width of the fishing gear, following the methods described by Hiddink et al. (2017).

To account for variations of the cumulative effect of demersal fishing disturbance on the seafloor, as well as the recovery of fishing grounds, additional fishing-related metadata attributes were extracted. For instance, time since fishing can be used to





assess the recovery of fishing grounds after its disturbance, and is often reported in experimental study designs as part of their

research objectives (Brylinsky et al., 1994; Falcão et al., 2003; Silveira et al., 2020). This variable can not only be used for the impacted site, but also for control sites. Several studies use areas that have been fished in the past but were not disturbed recently due to, for example, a temporary ban (Dannheim et al., 2014) or the recent establishment of an MPA (Brown et al., 2005; Sparks-McConkey and Watling, 2001) as their control sites, whereas some studies were able to identify areas that had never been fished as their control sites. To account for these contrasting control sites, we extracted the time since fishing, if

available, and created an additional column "Historically fished" that classifies whether a site had been fished in the past or not. To account for the cumulative impact of sediment disturbance (De Borger et al., 2021), we also extracted how much time had elapsed since the first disturbance (i.e. when fishing in the area had first begun), and if this information was not available, related bibliography describing historical fishing activity of the area was consulted.

Finally, the different response variables were harmonized, and complementary variables were calculated when possible (Table

3). For instance, although some studies provide a more precise classification of different sand fractions (e.g., fine sand, medium sand, Ramalho et al., 2018; Silveira et al., 2020), the majority of the studies simply provide total sand fractions (grain size > 63 µm) (Sciberras et al., 2017; Simboura et al., 2008), and these were calculated whenever possible.

**Table 3. List of harmonized response variables and harmonization method employed in DISOM.**

| Harmonized response variable | Harmonization method |
|---|---|
| Sand fraction | Sum of all sand fractions (e.g., fine sand, coarse sand) |
| Mud content (< 63 µm) | Sum of silt and clay fraction |
| Phytopigments | Sum of phaeopigment and chlorophyll concentrations |
| TOC/TN | Molar ratio between TOC and TN contents |
| Biopolymeric fraction of organic carbon | Sum of protein, lipid, and carbohydrate concentrations |

## 3 Database structure

Since each study analyses specific variables and reports different information, it would not be efficient to store all the data in a single spreadsheet. Therefore, a relational database was constructed to optimize the data storage capacity, where each table includes data from different categories that are linked together based on identifiers (i.e., primary and foreign keys such as study_id and sample_id) (Fig. 1). These different tables are: study_description, sampling_locations, and response_variables.

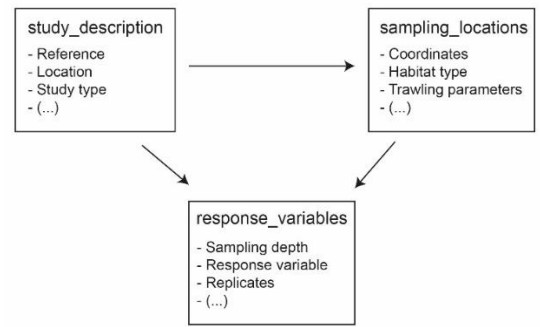

**Figure 1. Structure of the relational database.**

The first table, "study_description", stores information related to the publication, study area, and study design (Table S3). Since each publication can have several independent studies based on the study design or study area (see section 2.2), a unique identifier is given to each independent study.

The second table, "sampling_location", stores metadata associated to the sampling site such as its coordinates, sampling date, habitat type, seasonality, climate zone, and other variables related to fishing activity (e.g. fishing gear, fishing intensity, time since disturbance) which would help discern the effect of demersal fisheries (Table S4).

The third table, "response_variables", stores for each individual study and sampling location, the different sedimentological (i.e. grain size and porosity) and biogeochemical (i.e. OC, nitrogen, phytopigment, nutrient fluxes) variables measured, as well as the sample's metadata, such as sediment sectioning and how the data was collected (Table S5).

## 4 Database description

### 4.1 Spatiotemporal coverage of the data

The majority of the 71 studies included in the database were conducted in European waters, namely in the Mediterranean Sea (18) and in north European seas (e.g. North Sea, Irish Sea, Baltic Sea) (21) (Fig. 2a). Several studies were conducted in North America (15), mainly in the eastern margin, as well as along the Alaskan margin, the Gulf of Mexico, and the Gulf of California (Fig. 2a). Few studies were conducted in the Atlantic Iberian margin (4), south-eastern African margin (4), India (2), Brazil (3), New Zealand (2), Australia (1), and East Asia (1) (Fig. 1a). Despite the wide extension of fishing grounds along the southwestern South American and the east Asian margins, only 2 studies were identified in these extensively fished margins. Although demersal fishery occurs in all continental margins, we did not find peer-reviewed studies on trawling impacts and biogeochemistry along western South America and western Canada, nor along the western North African margin or the eastern Russian margin (Fig. 2a).

The first reported studies that assessed the impacts of bottom trawling on sediment biogeochemistry were published in the 1990s, with 5 independent studies during that decade (Fig. 2b). An increase in the number of studies was observed in the following decade, with 18 independent studies published between 2000 and 2010, whereas 35 studies were published during the last decade (2010-2020). In the last years (2020-2022), we have identified 13 studies that have addressed the impacts of demersal fisheries on biogeochemistry, which indicates a growing interest in understanding whether and how demersal fisheries may be altering biogeochemical cycles.



Earth System
Science
Data

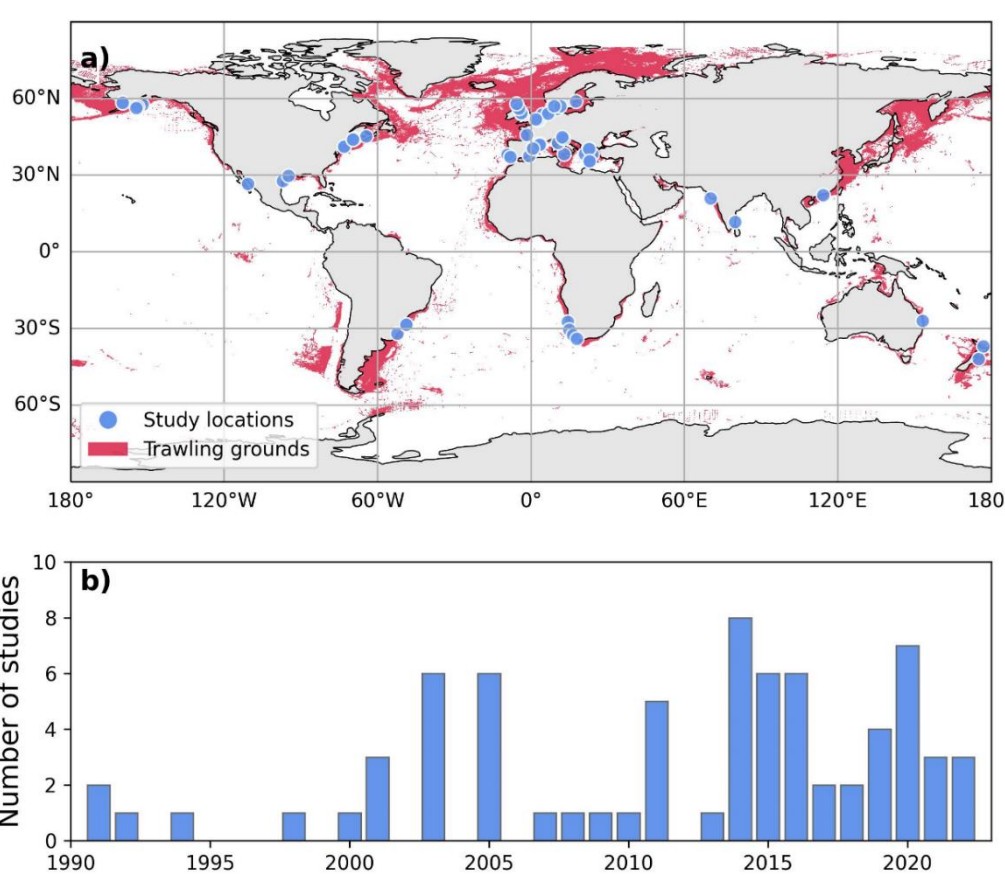

**Figure 2.** Spatial (a) and temporal (b) distribution of number of studies and publications that have addressed the impacts of demersal fishing on sediment biogeochemistry and sedimentology. Fishing grounds were extracted from Global Fishing Watch v.2.0 (Kroodsma et al., 2018). Note that Global Fishing Watch does not distinguish between demersal and pelagic trawling, so this spatial extension should be used informatively.

### 4.2 Distribution of study designs

The distribution of study designs stored in DISOM is relatively equally distributed, with a similar representation of experimental (25), comparative control-impact (20), and comparative gradient (26) studies (Fig. 3). Studies conducted on the European continental margins include an equal number of the three study designs (Fig. 3). In North America, experimental and comparative control-impact studies have been conducted equally, but no comparative gradient studies were carried out (Fig. 3). Studies conducted in South America, Africa, Asia, and Oceania, only presented one typology of study design (Fig. 3). Future studies should be carried out with all of the 3 typologies of study designs in order to deepen the knowledge of acute and chronic impacts and the effects of different fishing intensity on organic matter in these regions.



**Figure 3. Spatial distribution of study designs on global continental margins (a), and on European margins (b) in DISOM. Summary of the different study designs in the dataset (c).**

## 4.3 Distribution of environmental conditions

Studies of the effects of different bottom fishing gear in different habitat types on sediment biogeochemistry are essential for an understanding of fishing impacts using global models. The number of independent studies in sandy and muddy habitats is equally represented in DISOM, with 37 and 34 independent studies in sandy and muddy habitat types, respectively (Fig. 4). However, except for the European and American margin, there is an unequal spatial distribution of the studies on sandy and muddy habitats: Brazilian and southwest African margins only presented studies on sandy environments whereas Chinese, Australian and New Zealand margins only had studies on muddy seafloors (Fig. 4).



Earth System Discussions
Science
Data
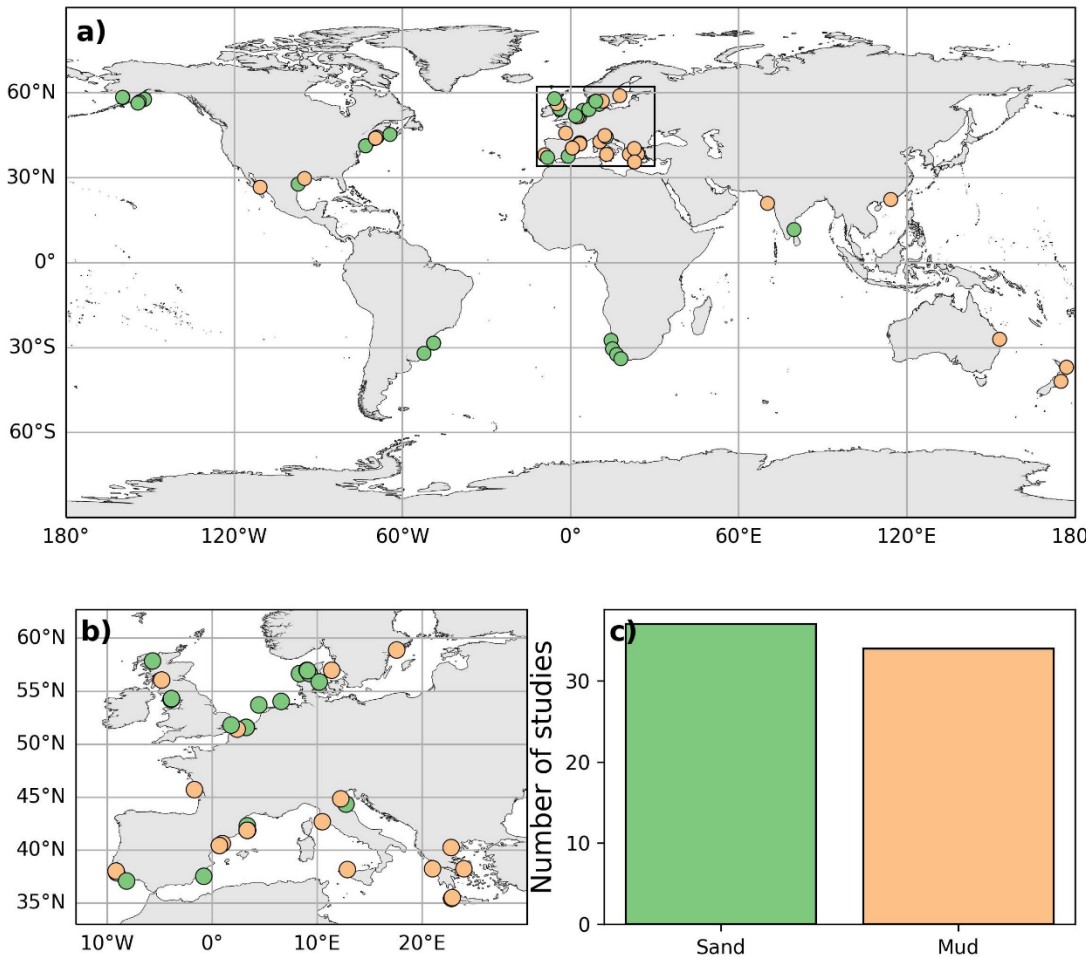


**Figure 4. Spatial distribution of habitat types on global continental margins (a), and on European margins (b) in DISOM. Summary of the different habitat types in the dataset (c).**

The majority of studies were conducted in shallow environments, at less than 50 m depth, whereas few studies were conducted at deeper environments (>200 m) (Fig. 5). The highly skewed nature of the dataset towards shallow environments should be

considered when modelling the effects of demersal fisheries, especially considering the continuous expansion of fishing grounds to deeper environments (Norse et al., 2012; Watson and Morato, 2013).



**Figure 5. Spatial distribution of habitat types on global continental margins (a), and on European margins (b) in DISOM. Summary of the different habitat types in the dataset (c). Note that the colorbar is centred at 130 m water depth to highlight the variations in**
**sampling depths.**

Finally, one of the factors that should be considered when studying the impacts of demersal fisheries is the sampling season, since the impacts caused by this anthropogenic activity could be masked if they coexist with seasonal storms or cycles of high net primary productivity. For instance, seasonal storms can resuspend as much sediment and particulate organic carbon as bottom trawling (Arjona-Camas et al., 2021, 2022; Durrieu de Madron et al., 2005; Ferré et al., 2008; Mengual et al., 2016;
Paradis et al., 2022; Pusceddu et al., 2005b, 2015), whereas seasonal variations in organic matter inputs to the seafloor could offset the effects of trawling (Daly et al., 2018; Rajesh et al., 2019; Smith et al., 2000). Although several studies conducted their sampling during several seasons to account for the effect of seasonality (20), the majority of the studies included in





DISOM only sampled during a specific season, usually during summer (30), which could skew the global results when assessing the impacts of demersal fisheries on sediment biogeochemistry (Fig. 6).

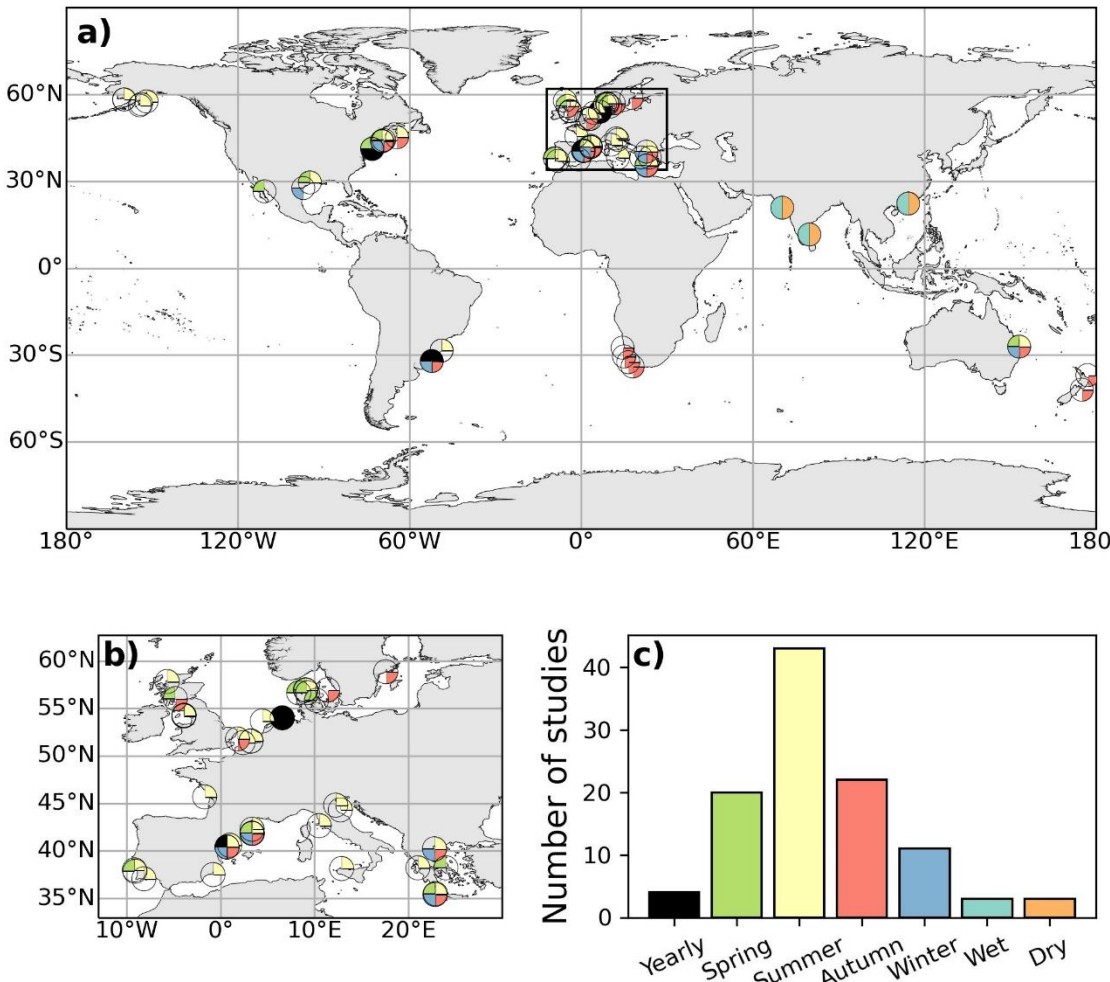


**Figure 6. Spatial distribution of sampled seasons (filled quadrants) or whether data collected during different seasons was pooled together yearly (black circles) on global continental margins (a), and on European margins (b) in DISOM. Summary of the different seasons sampled in the dataset (c).**

**4.4 Distribution of fishing descriptors**

Although we harmonized the different fishing gear types, it is important to note that there is a wide variety of configurations for each gear type with different effects on the seafloor (Martín et al., 2014a). For instance, the size and weight of otter trawls increases with fishing ground depth, so an otter trawl that operates in shallow environments will be different than one that operates at greater depths (Ragnarsson and Steingrimsson, 2003).

In this dataset, there was an unequal distribution of demersal fishing gear types, which could skew statistical analyses. The
most common fishing gear type identified was otter trawls (39), followed by towed dredges (18) and beam trawls (7) (Fig. 7).



The type fishing gear was sometimes undefined in certain studies (7). In these situations, demersal fishing fleets consisted of a mixture of otter trawl, dredges and beam trawls (Fig. 7). The effects of electric pulse trawls and artisanal gears were only assessed by one study each, and these were spatially limited: the study on electric pulse trawl technique was only conducted in the North Sea, while artisanal gears were only studied in southern Brazil (Fig. 7).

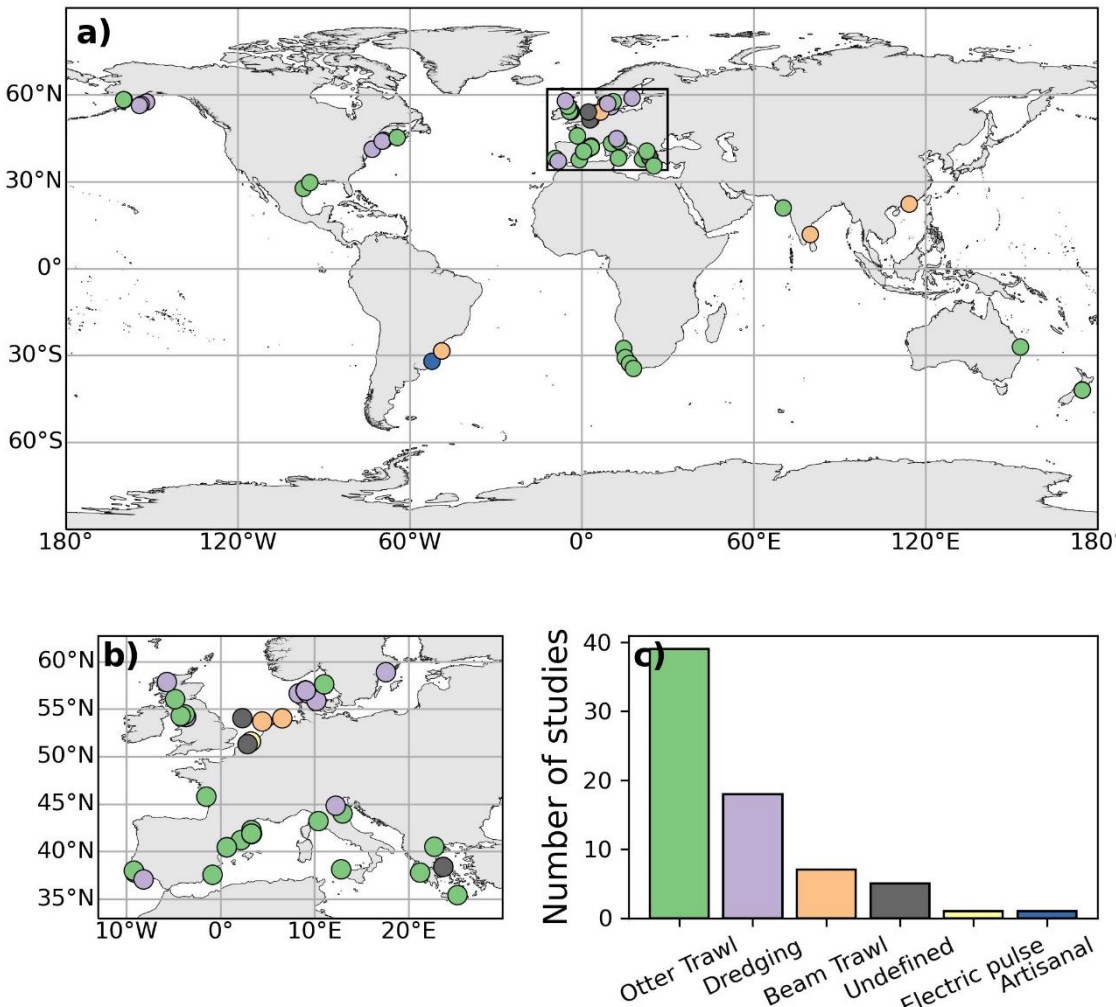


**Figure 7. Spatial distribution of fishing gear type used for each study on global continental margins (a), and on European margins (b) in DISOM. Summary of the different fishing gear type in the dataset (c).**

From the 71 independent studies, ~75 % did not have a true control site (i.e. where bottom fishing was completely absent) to compare the effect of this anthropogenic disturbance in relation to natural baseline conditions (Fig. 8). These studies either did
not have any control site (no control) or their control site had been disturbed in the past (historically fished) and may not have recovered completely to its baseline conditions.





Studies with no true control site were either experimental studies that sampled before and after disturbance in an impacted site (Bhagirathan et al., 2010; Tiano et al., 2021), or were comparative gradient studies that sampled in areas with different fishing intensities (Atkinson et al., 2011; Hale et al., 2017; Sciberras et al., 2016). Considering that demersal fishing grounds

essentially cover all continental margins, it is not surprising that a representative control site with similar environmental conditions as the disturbed site is most often not available (Fig. 8). Indeed, only 17 independent studies had true control sites due to specific conditions: i) experimental studies were conducted in marine protected areas that had never been fished (Ferguson et al., 2020; Morys et al., 2021; Tuck et al., 1998), ii) control sites were far from harbours and fishermen still had not explored those areas (Paradis et al., 2019; Simboura et al., 2008; Watling et al., 2001), or iii) because specific features

prevented the access of fishing vessels to certain areas (Cartes et al., 2007; Martín et al., 2014b; Palanques et al., 2014; Paradis et al., 2021).

To overcome this lack of true controls, certain studies chose sites that were not currently disturbed, but had been historically fished as pseudo-control sites ("Historically fished" in Fig. 8). These sites were not being disturbed during the study because they were conducted during a seasonal fishing closure (Fiordelmondo et al., 2003; Meseck et al., 2014; Polymenakou et al.,

2005; Pusceddu et al., 2005a; Silveira et al., 2020; Tiano et al., 2019), or because the site had been recently banned from fishing activities (Brown et al., 2005; Dannheim et al., 2014). However, since these pseudo-controls had been disturbed by demersal fisheries in the past, they may not have recovered completely to its baseline conditions, which could affect the outcomes of the study. For instance, a recent global meta-analysis on the effects of demersal fisheries on benthic communities revealed that the history of fishing disturbance prior to an experimental study influenced the outcome of the result, masking

the impacts in comparison to studies on undisturbed sites (Sciberras et al., 2018).

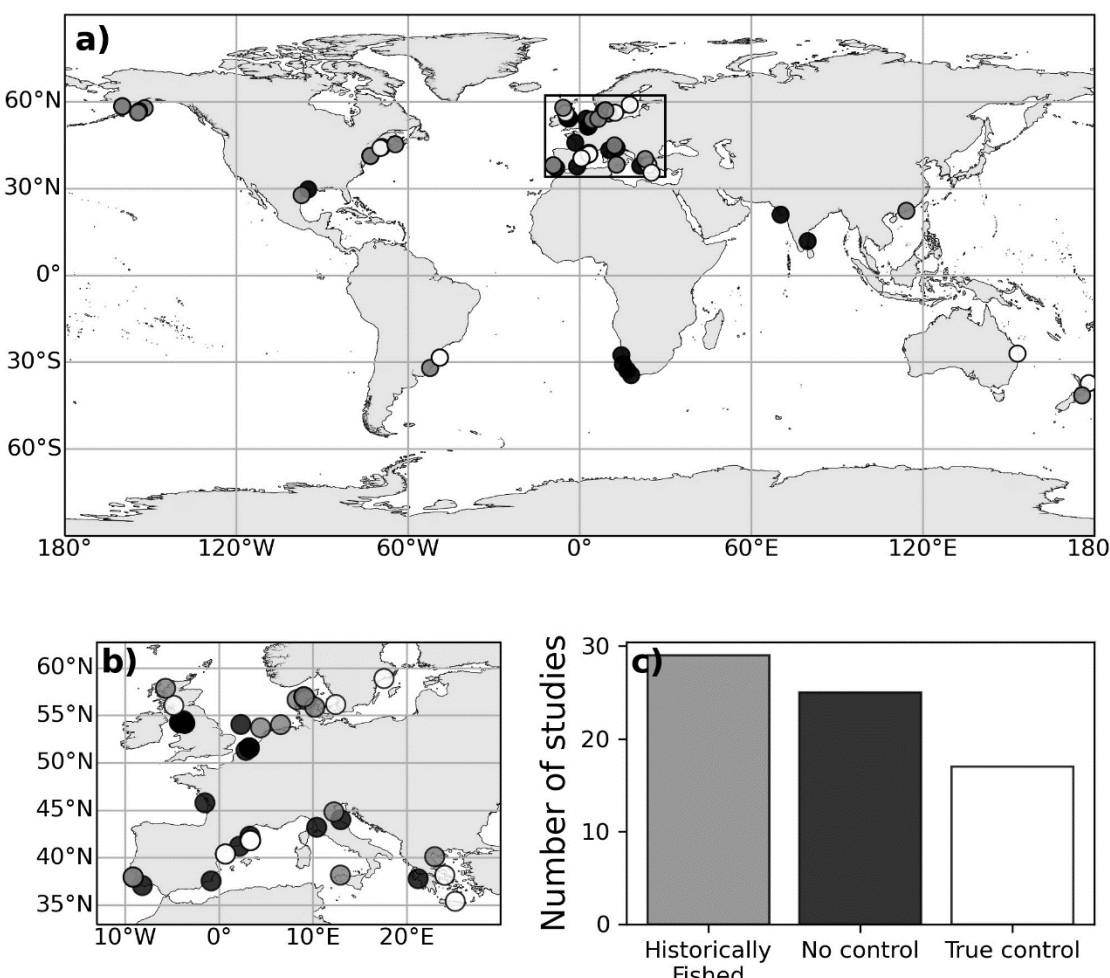

**Figure 8. Spatial distribution of presence of a control site for each study on global continental margins (a), and on European margins (b) in DISOM. Summary of the different types of controls in the dataset (c).**

**4.5 Response variables**

We identified 90 different sedimentological and biogeochemical variables reported in all studies, and the 10 most reported ones are shown in Figure 9. These are separated based on the different study designs (experimental, comparative control-impact, comparative gradient) and the type of information they provide (sedimentological and biogeochemical). Unsurprisingly, the most widely measured response variable for all study designs was organic carbon (OC) (Fig. 9). Total nitrogen and proxies of algal material (e.g. chlorophyll-a, phaeopigments, and the total phytopigment concentration), which

provide a useful measure of the fishing effect on the nutritional quality of organic matter (i.e. its food availability for the benthos; Pusceddu et al., 2009), were commonly reported in both experimental and comparative control-impact study designs (Fig. 9). However, other organic compounds such as proteins, lipids and lignin phenols were not sufficiently represented across

studies, hindering the capacity to study the effects of demersal fisheries on different organic compounds with different

reactivities and origins.

The effect of sediment disturbance on sedimentological variables, such as grain size, was represented in all study designs, but

these variables were more prominent in comparative gradient studies.

Biogeochemical variables which measure remineralization processes (e.g. nutrient concentrations and fluxes, as well as oxygen

consumption) were more commonly reported in manipulative and comparative gradient studies than in comparative control-

impact ones (Fig. 9). The unbalanced representation of these variables among study designs may bias our understanding of the

effect of fishing disturbance on remineralization processes in marine sediments, and more studies that assess the effect on

organic matter remineralization should be performed.

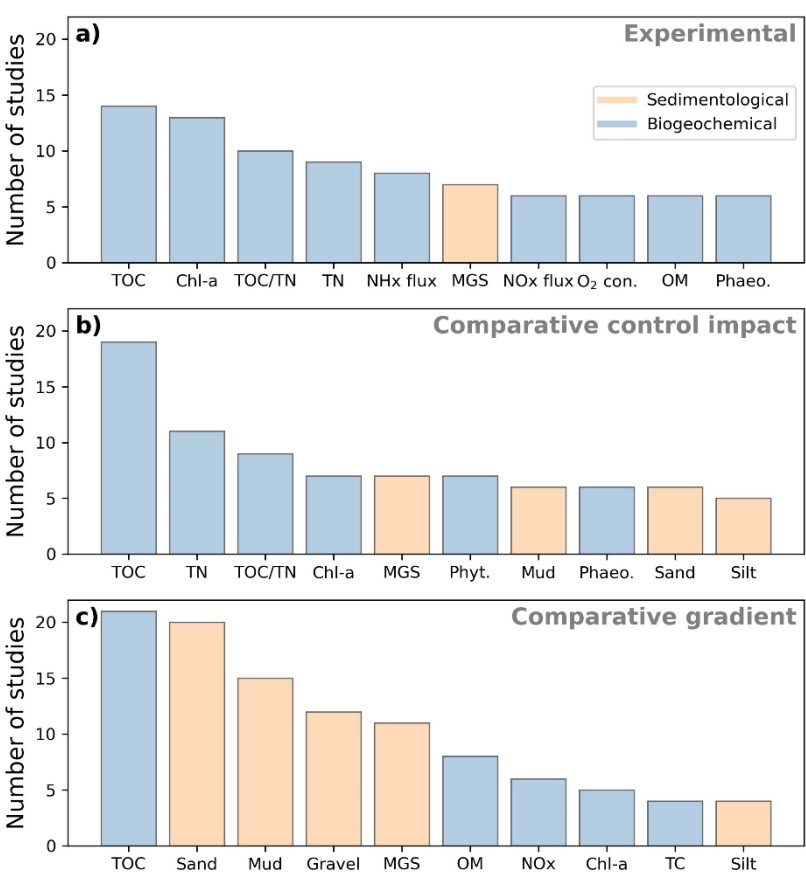

**Figure 9. Number of studies that analysed each response variable in each of the three study designs: (a) experimental, (b) comparative control-impact, (c) comparative gradient studies. For clarity, only the 10 most common response variables were**
**included, and variables were grouped based on the property they represent (sedimentological or biogeochemical). A full description of the different biogeochemical and sedimentological variables can be found in Table S2.**





### 4.6 General remarks

Each individual study in this compilation has a unique combination of environmental conditions (e.g., habitats, depths, and seasonality) and demersal fishing descriptors (e.g., fishing gear type, presence of control site) (Fig. 10), which need to be considered when comparing their results and providing a greater perspective of the effect of demersal fishing disturbance on the seafloor. For instance, the majority of the comparative gradient studies did not have any control site given the difficulty of finding a representative control site in the area under scrutiny. Given their nature, beam trawls were almost exclusively used in sandy environments, and no comparative gradient study was performed using this gear.

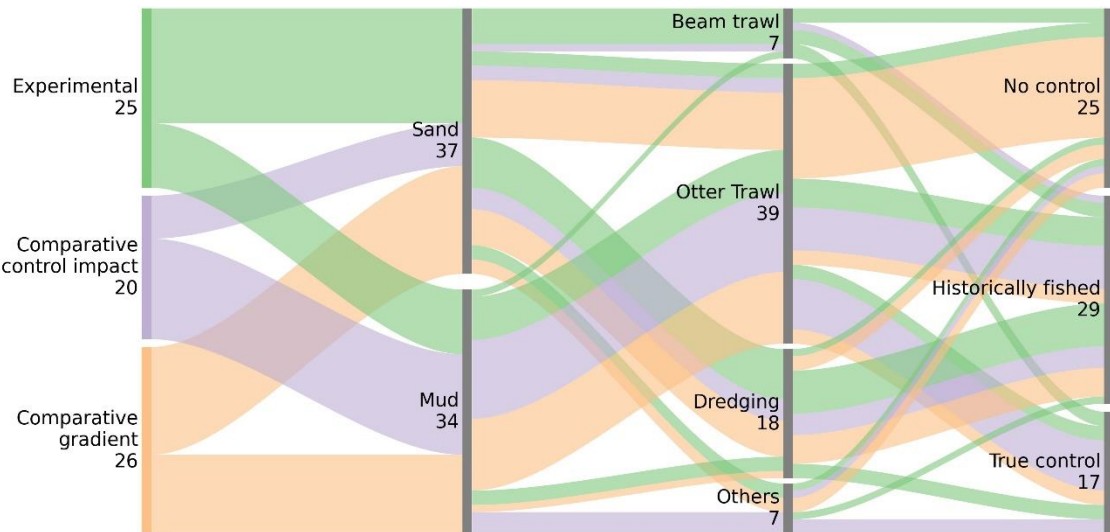

**Figure 10. Sankey diagram of the different study designs, sampling strategy, and fishing descriptors found in the literature, highlighting the complexity of the dataset. Colours represent the study design.**

These unique combinations of study design, environmental conditions, and fishing descriptors may limit what is currently known on the effect of demersal fishery on biogeochemical processes on the seafloor, producing contrasting results in their biogeochemical effects on the seafloor (Epstein et al., 2022). The reasons behind these contrasting results are beyond the scope of this manuscript and should be addressed in a thorough meta-analysis.

While this database only gathered data from studies published in peer-reviewed manuscripts, there is a wealth of information available in grey literature that could further improve our understanding of the effect of demersal fisheries on global biogeochemical processes in marine sediments. Moreover, future versions of this database should incorporate data of benthic community abundance and composition, since alterations of benthic communities can also affect the biogeochemical and sedimentological properties of the seafloor. Finally, considering that demersal fisheries modify sedimentary dynamics and the composition of suspended sediment and nutrients in the overlying water column, these effects should be included to properly understand the global effect of demersal fisheries on biogeochemical cycles in the marine realm. As the number of these studies increases, DISOM should be expanded to include studies that address the effect of demersal fisheries on the water column.

**5 Data availability**

The database is available for download as an Excel file, where each separate sheet stores the individual tables (Paradis, 2023; https://doi.org/10.3929/ethz-b-000634336).

**6 Conclusion**

DISOM is the first comprehensive open access database that compiles data of sedimentological and biogeochemical variables of peer-reviewed studies that address the effects of demersal fisheries on the seafloor. We also present important metadata on
environmental and fishing descriptors that need to be reported to ensure the comparability between studies and facilitate holistic meta-analyses. While this database harmonizes data from 71 independent studies located in a wide variety of continental margins, we have identified gaps along the African, South American, Canadian, and Asian margins. More studies should be conducted in these margins to account for the effect of sediment disturbance in all environmental conditions and understand the effects of demersal fishing on global biogeochemical processes in marine sediments.

**Author contribution**

SP performed the first screening step while all authors performed the second screening step. All authors devised a methodology to objectively extract the data, and compiled the data of assigned manuscripts. SP harmonized the data across studies. SP wrote the manuscript with the contribution of all co-authors.

**Competing interest**

None of the authors have any competing interests.

**Acknowledgements**

We would like to acknowledge Albert Palanques, Ciarán McLaverty, Rachel Hale, Irini Tsikopoulou, Mark Trimmer, Christopher J. Smith, and Lawrence M. Mayer who provided data in tabular format when they weren't available in the publications.

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
