# Peer review of "Demersal fishery Impacts on Sedimentary Organic Matter (DISOM): A global harmonized database of studies assessing the impacts of demersal fisheries on sediment biogeochemistry"

_Earth System Science Data, 2023_

## Author Response (AR1)

**RC1:**

Paradis and co-workers have synthesized the data related to the impact of trawling on the sedimentological characteristics and organic carbon burial in the global marine sediments. As stated by the authors, bottom trawling severely impacts the ocean community. Unfortunately, the methodology followed to assess the impact of trawling on marine community varies case-to-case basis. Therefore, a uniform comprehensive methodology to assess the harmful impacts of trawling on benthic community will help tremendously. However, it is difficult to suggest a common methodology for all regions, due to the huge diversity in sediments and environments. This assessment suffers from the lack of coherency in different studies.

We would like to thank the reviewer for their time reviewing this manuscript and for acknowledging the importance of harmonizing data from different studies that assessed the biogeochemical impacts of demersal fisheries.

The lack of coherency between different studies (in terms of methodology and environmental settings) and inconsistency in results is one of the main reasons we saw the need for a quality-controlled, harmonized dataset that can be used to improve the robustness of future quantitative analyses. We discuss this rationale in the Introduction and have added further comments throughout the text, as outlined in our detailed comments below.

Major Concerns

- Many previous studies have not included the details of fishing frequency properly. How does this lack of information about the fishing frequency in several studies affect the outcome of this synthesis?

Fishing effort is reported differently in each study. To harmonize this information, we converted it to fishing frequency following the methods described by Hiddink et al. (2017). Despite the importance of knowing the fishing effort to be able to relate the magnitude of the observed impacts, 26 out of the 71 studies included in DISOM do not provide this information. The majority of these studies were comparative control-impact studies that collect sediment cores in a known fishing ground and in a control site. Since the aim of these studies is solely the comparison of the biogeochemical properties between impacted and non-impacted sites, they seldomly collect this information. In addition, many of these studies were conducted prior to the installation of geographical positioning systems in fishing vessels, so the estimation of fishing frequency relied on the use of logbooks, which were not always available. However, with the installation of GPS in fishing vessels that are then broadcasted as AIS and VMS data, the estimation of fishing effort is becoming more available.

We have included the description of the amount of studies that do not provide any measurement of fishing effort in the revised manuscript in the section 4.4 Distribution of fishing descriptors, which reads:

> "Finally, only 45 studies provided information of fishing frequency, which eventually will limit the possibility to use these data to fully understand how fishing frequency affects the magnitude of the sedimentological or biogeochemical parameter measured. The majority of the 26 studies that did not include fishing frequency are comparative control-impact studies that designed their studies to compare the

biogeochemical properties between impacted and control sites, without considering the influence of fishing frequency. In addition, many of these studies were conducted prior to the installation of geographical positioning systems in fishing vessels, such as Automatic Identification Systems (AIS) or Vessel Monitoring Systems (VMS), that facilitate the estimation of fishing frequency over an area. With the increasing availability of harmonized fishing frequency data such as through the efforts of Global Fishing Watch, we will be able to include this information without the need to harmonize fishing effort."

In addition, we have included how the limited availability of this metric would complicate the global assessment of the biogeochemical impacts of demersal fisheries in the section 4.6 General remarks (see below for additional changes in this section).

- The fishing gear used in synthesized studies varies a lot, again hampering a proper assessment of the fishing impact of stated sediment variables.

As the reviewer well notes, the fishing gear type used can influence the biogeochemical effect observed. The broad variety of fishing gears complicate a proper assessment of the impacts of demersal fisheries by reducing the degrees of freedom if we want to account for the effects of fishing gear types. We have included a small discussion of this issue in the section 4.6 General remarks (see below for additional changes in this section).

- Given the incoherency in the synthesized studies, is it really possible to use the compiled data to model the fishing impacts with reasonable accuracy?

Indeed, the heterogeneity of variables compiled in this dataset complicates the modelling of the fishing impacts. We highlight these heterogeneities in Figure 10 so that researchers that aim on using this database to model the impacts of demersal fisheries are aware that they need to take into account that studies used, for instance, different sampling designs, depths, or seasons, as well as different fishing descriptors such as fishing gear type, presence of a true control, and fishing effort, all of which could be influential variables. In essence, there are many variables that can affect the magnitude of the effects observed, whereas there are limited number of studies available (71), reducing the degrees of freedom if we want to perform statistical analyses.

This is now stated in the following sentence in the amended manuscript, in the section 4.6 General remarks:

> "Hence, researchers that aim to use this database should take into account that the heterogeneity of the large number of influential variables (i.e., study design, environmental conditions, fishing descriptors) will reduce the degrees of freedom needed to conduct meaningful statistical analyses."

- It's hard to assess the applicability of the compiled descriptors with huge variation from study to study. Isn't it better to just put this work as a review article, rather than a data descriptor?

We believe that in order to advance the science of demersal fishing impacts a quantitative (e.g. meta-analysis) rather than descriptive (review) approach is needed. We present here a compilation of data extracted from 71 independent studies that amount to 907 independent

sampling locations, with the aim to then use this information to properly disentangle the reasons behind the contrasting biogeochemical impacts of demersal fisheries on the seafloor. To do so, we require that the data is harmonized between studies, to ensure their comparability. Hence, the aim of this data descriptive manuscript is to provide a harmonized dataset to allow other researchers to use it to conduct meta-analyses or other statistical analyses. Using this data, several meta-analyses could be performed depending on their objectives, which is why we wanted to make the data available for other researchers to perform their analyses as they would see fit.

- I could not access the dataset, as it asked for certain details and login.

  While the database is currently under embargo until the publication of a meta-analysis, it was made accessible to all reviewers, as requested by the editors of ESSD.

  Minor Comments

- Line 28, 'increasing in intensity'
- Line 33, replace 'has been shown to change' with 'changes'
- Line 37, 'increase'
- Line 39, 'increase', 'decrease'
- Line 47, what is meant by 'very hydrodynamic area'? An area that has strong bottom current intensities, which has been included in the manuscript
- Line 205, replace 'for an understanding of fishing impacts' with 'to understand the fishing impacts'
- Line 213, replace 'less then' with '<'

  Line 241, insert 'of' before 'fishing'

  These changes have been included in the revised manuscript.

**RC2:**

Paradis et al., has brought together data related to the impact of bottom trawling impact on a number of biogeochemical and sedimentological variable's from across the global oceans. Currently, the types of data collected and the way it was collected differs significantly, making global assessments of the impact on sediments. This new uniform dataset will lay the foundations for regional and global scale assessments of trawling impacts on the seabed. The manuscript itself will be of interest a wide audience and is of high quality with the exception of a few minor graphical components.

For review it would have been useful to see the dataset, but this should not hamper the publication of this work.

While the database is currently under embargo until the publication of a meta-analysis, it was made accessible to all reviewers, as requested by the editors of ESSD.

Figure 6 and 7 box C - the x-axis titles don't align with the ticks.

The center of the x labels of Figures 6c and 7c aligns with their ticks.

Figure 10 - this is very complex and hard to understand is there another way to present this data or do you need this plot

Figure 10 is a Sankey diagram of how the different study designs were performed on different habitat types and using different fishing descriptors, highlighting that the studies in DISOM present a unique combination of study design, environmental conditions and fishing descriptors, all of which could influence the biogeochemical impacts of demersal fisheries. This unique combination of variables could explain the contrasting biogeochemical impacts of demersal fisheries observed in the different studies. As explained to Reviewer 1, this heterogeneity of the large number of influential variables decreases the degrees of freedom when performing statistical analyses, which is why we want to highlight this with the Sankey diagram of Figure 10.

To clarify Figure 10, we have included the following sentence in the amended manuscript:

"As illustrated in the Sankey diagram (Fig. 10), each individual study in this compilation has a unique combination of environmental conditions (e.g., habitats, depths, and seasonality) and demersal fishing descriptors (e.g., fishing gear type, presence of control site, information on fishing effort), which need to be considered when comparing their results and providing a greater perspective of the effect of demersal fishing disturbance on the seafloor."

I have not had time to look through and list typos.

**RC3**

Review: Demersal fishery Impacts on Sedimentary Organic Matter (DISOM): A global harmonized database of studies assessing the impacts of demersal fisheries on sediment biogeochemistry.

General.

Bottom trawling is one of the dominant anthropogenic activities that disturb the seafloor and benthic ecosystem. The study of its impact has mainly focused on the structure of the sediment and on marine life, in particular the damage / mortality of benthic organisms and impact on the food-web. More recently, concerns were raised about the impact of bottom trawling in relation to global warming as the disturbance of the ocean floor by bottom trawling may have implications for the sequestering of organic carbon. The current paper makes an important contribution to the scientific literature as it collates and reviews the peer reviewed publications on field experiments on the effects of bottom trawling on the sediment biochemistry. The authors create a data base of the published results which will facilitate meta-analysis of the data. I could not access the data base but only the excel spreadsheet of the supplementary material, which provides information on the data reported by the reviewed papers. The entries scored provides important information on the for instance the biogeochemical parameters, seafloor habitats and geographic regions which are not well represented in the literature. As such the paper provides important guidance for future studies to contribute input for future meta-analysis.

The authors distinguish between broad gear categories, each encompassing a wide range of specific gears that differ in their impact on the sediment / biogeochemistry. I believe that it would be helpful if they could include some recommendations on the gear parameters that could improve the quantification of the impact.

The reviewer points out an important aspect regarding the general classification of fishing gear types. While otter trawling is the most common fishing gear type, it can have several configurations that result in different levels of impact on the seafloor.

We have added the following sentence in the section 2.2 Data harmonization:

"While this classification groups together gears with similar properties, it may mask the nuances of the degree of impact each specific fishing gear could have. Hence, we advise future studies to specify, in addition to this general gear type, the model, dimensions and weight of the gear type as well as the targeted commercial species, all of which could affect the magnitude of the impacts."

The paper is well written. I have only a few minor comments which the authors could consider in revising their text. I recommend that the paper can be published taking the minor comments into consideration.

Specific comments

Line 41-42. Wondered why the authors did not refer to the recent critical comment on the Sala paper. Hiddink et al. 2023 https://doi.org/10.1038/s41586-023-06014-7.

We have included this relevant and critical rebuttal in the amended version of the manuscript.

Line 177. extensively might be misunderstood. do you mean that bottom trawling is widespread in these margins, or that the fishing intensity is generally low.

"Extensively" was modified to "widespread" to highlight that these regions have large fishing grounds.

**RC4**

Manuscript entitled: Demersal fishery Impacts on Sedimentary Organic Matter (DISOM): A global harmonized database of studies assessing the impacts of demersal fisheries on sediment biogeochemistry. The following manuscript discriminates the demersal fishery and its impact on the biogeochemistry of the sea floor sediments. It is considered as the main anthropologic activity affecting the shallow marine biological system and sedimentation types and rates. The manuscript is well organized and well written and the produced dataset is very helpful in assessments of demersal fishery and their impact on the sea floor on regional and global. Some modifications are required in order to be suitable for publication.

1)   However great effort in data completion and filtering, more clarification about how the authors harmonize the different incomputable data from the published manuscripts; different gear types, different lithological measurements, biochemical parameters …etc for each different area.

We describe how the different incomputable variables are harmonized between studies:

Lithological habitat type:

> "Two different habitat types were defined based on the sedimentological properties of the sites: "muddy" if the percentage of mud (sediment grain size < 63 µm) was higher than that of sand, or "sandy" if the percentage of sand was higher than mud. If this information was not available, we extracted this classification based on the manuscript's description, or from associated bibliography of the study area."

Seasonality:

> "Seasons varied throughout the different climatic zones encompassed in this database. Although local variations in seasons may exist, temperate zones are usually characterized by four seasons (winter, spring, summer, autumn) and tropical zones usually have two main contrasting seasons (wet and dry). The dataset was harmonized into these different categories."

Gear types:

> "Fishing gear type was classified into six main categories: otter trawls, beam trawls, dredges, electric pulse trawl, artisanal fishing gears, and undefined fishing gears (Table 2). If a study did not specify what type of fishing technique was used, it was extracted from other studies or technical reports from the area, and if this information could not be obtained anywhere, it was assigned as "undefined"."

With respect to figures, they need some modifications as follows;

Figure 6 the circles represent different seasons are not clearly observed (needs to be minimized) and map itself needs to be enlarged.

To be consistent, the size of the map of this figure is the same as the other figures presented in this manuscript. However, we acknowledge that the labels of the seasons may be difficult to interpret. We have improved its aesthetics to facilitate its interpretation.

Figure 10 is very complicated and in-illustrative, need clarification.

Same reply to Reviewer 2 on this figure:

Figure 10 is a Sankey diagram of how the different study designs were performed on different habitat types and using different fishing descriptors, highlighting that the studies in DISOM present a unique combination of study design, environmental conditions and fishing descriptors, all of which could influence the biogeochemical impacts of demersal fisheries. This unique combination of variables could explain the contrasting biogeochemical impacts of demersal fisheries observed in the different studies. As explained to Reviewer 1, this heterogeneity of the large number of influential variables decreases the degrees of freedom when performing statistical analyses, which is why we want to highlight this with the Sankey diagram of Figure 10.

To clarify Figure 10, we have included the following sentence in the amended manuscript:

> "As illustrated in the Sankey diagram (Fig. 10), each individual study in this compilation has a unique combination of environmental conditions (e.g., habitats, depths, and seasonality) and demersal fishing descriptors (e.g., fishing gear type, presence of control site, information on fishing effort), which need to be considered when comparing their results and providing a greater perspective of the effect of demersal fishing disturbance on the seafloor."